# The Protective Role of TREM2 in the Heterogenous Population of Macrophages during Post-Myocardial Infarction Inflammation

**DOI:** 10.3390/ijms24065556

**Published:** 2023-03-14

**Authors:** Sang Hyun Kim, Kwan Yong Lee, Kiyuk Chang

**Affiliations:** 1Division of Cardiology, Department of Internal Medicine, Seoul St. Mary’s Hospital, College of Medicine, The Catholic University of Korea, Seoul 06591, Republic of Korea; 2Cardiovascular Research Institute for Intractable Disease, College of Medicine, The Catholic University of Korea, Seoul 06591, Republic of Korea

**Keywords:** myocardial infarction, macrophage, TREM2, monocyte, protective inflammatory

## Abstract

Advances in interventions after myocardial infarction (MI) have dramatically increased survival, but MI remains the leading cause of heart failure due to maladaptive ventricular remodeling following ischemic damage. Inflammation is crucial in both the initial response to ischemia and subsequent wound healing in the myocardium. To date, preclinical and clinical efforts have been made to elucidate the deleterious effects of immune cells contributing to ventricular remodeling and to identify therapeutic molecular targets. The conventional concept classifies macrophages or monocytes into dichotomous populations, while recent studies support their diverse subpopulations and spatiotemporal dynamicity. The single-cell and spatial transcriptomic landscapes of macrophages in infarcted hearts successfully revealed the heterogeneity of cell types and their subpopulations post-MI. Among them, subsets of Trem2^hi^ macrophages were identified that were recruited to infarcted myocardial tissue in the subacute phase of MI. The upregulation of anti-inflammatory genes was observed in Trem2^hi^ macrophages, and an in vivo injection of soluble Trem2 during the subacute phase of MI significantly improved myocardial function and the remodeling of infarcted mice hearts, suggesting the potential therapeutic role of Trem2 in LV remodeling. Further investigation of the reparative role of Trem2 in LV remodeling would provide novel therapeutic targets for MI.

## 1. Introduction

Survival and the clinical outcomes of acute myocardial infarction (MI) have improved over the years, owing to major advances in cardiac revascularization strategies and the introduction of various pharmaceutical interventions [1]. However, heart failure through adverse left ventricular (LV) remodeling is a common sequela in MI survivors and causes a significant number of morbidities and mortalities in MI survivors [2,3].

After myocardial infarction, ventricular remodeling or recurrent ischemic events are both reported to be associated with inflammation [4]. In particular, during the early phase of cardiac injury, inflammation plays a crucial role in determining the extent of myocardial injury, including cardiomyocyte necrosis and the degradation of extracellular matrix components. Prior clinical studies investigating the use of existing anti-inflammatory agents to treat MI have provided important insights to investigators. These studies have revealed that the pathogenesis of cardiovascular disease involves a much more complicated process of inflammation, and targeting non-specific steps of inflammation may not influence clinical outcomes [5,6]. A more specific site and a cellular subset targeting intervention that could mitigate inflammation without interfering with natural myocardial healing are needed. Promoting the protective inflammatory reaction involving the activation of specialized subsets of immune cells, such as macrophages and neutrophils, could be an alternative.

Specifically, macrophages have been suggested to play a pivotal role in the clearance of infarcted tissues, as well as in wound repair and remodeling processes [7]. Subsets of macrophages have been sorted and investigated for their molecular spectrum and role in immune responses. Traditionally, they are broadly divided into M1 and M2 subtypes according to in vitro construction; M1 macrophages promote a proinflammatory environment with high levels of proinflammatory cytokines, whereas M2 macrophages promote angiogenesis and wound healing with anti-inflammatory cytokines [8]. While this dichotomous classification and their balancing effects may explain the healing and remodeling of the myocardium in a conceptual manner, the in vivo environment and dynamic interactions of macrophages are much more complex. Therefore, today’s great challenge is to accurately characterize the dynamics of macrophages during the acute period of MI. Recent studies using new techniques, such as single-cell RNA sequencing (scRNA-seq), have revealed the spatiotemporal relationship and function of various macrophage subsets in cardiac remodeling after myocardial infarction [9,10].

In this article, we review major studies to answer these long-standing questions to investigate these underlying inflammatory processes and the spatiotemporal dynamics of MI-associated immune cells. In particular, we focus on the role of Trem2^hi^ macrophages and the soluble form of triggering receptor expressed on myeloid cells 2 (Trem2), dominant anti-inflammatory immune cells in infarcted myocardium and its key molecule, leading to significant functional and structural improvements in infarcted hearts.

## 2. Role of Inflammation in Myocardial Infarction

LV remodeling is a process in which the heart adapts to mechanical, neurohormonal, and inherited changes by regulating the size, shape, and function of the ventricle. It is commonly observed in infarcted hearts [4]. Adverse LV remodeling following MI involves various processes, including the transformation of LV geometry, myocardial fibrosis, a shift in energy metabolism, neurohormonal activation, and inflammation. A plethora of evidence supports the importance of an orchestrated immune response in injured cardiac myocytes via the infiltration of neutrophils, macrophages, or T cells. Major shifts in the microenvironment of cardiomyocytes and the deposition of scar tissue through fibrosis are well-known, and they are also introduced in the concept of the reverse remodeling of LV as a healing process [11].

Inflammation has been suggested to be a classic pathogenesis that plays a key role in atherosclerosis, both in the early phase of initiation and the long-term progression of vulnerable plaques [12]. The JUPITER trial has shown better clinical outcomes of statins in primary prevention due to their anti-inflammatory properties, independent of their low-density lipoprotein-lowering effect [13]. Some key properties of statins may contribute to plaque stabilization by reducing inflammatory cell adhesion and monocyte recruitment to endothelial cells, changing smooth muscle migration in developing plaques and favorably affecting matrix metalloproteinases.

A number of clinical studies have been conducted on MI models to identify inflammatory molecules or responsible immune cells and, ultimately, to find novel therapeutic targets. The CANTOS trial demonstrated not only that increased serum levels of high-sensitivity C-reactive protein (CRP) and interleukin (IL)-6 were associated with elevated cardiovascular risks but also that the blockade of IL-1 beta with a monoclonal antibody in patients with a history of MI and an elevated baseline level of CRP was followed by significantly lowered all-cause and cardiovascular mortalities [14]. This study provided evidence that modulating certain steps of inflammation may influence the pathogenesis of cardiovascular disease and encouraged investigators to find alternative anti-inflammatory interventions to mitigate inflammation. In subsequent randomized clinical trials, colchicine was shown to significantly improve ischemic cardiovascular outcomes in both recent MI and chronic coronary disease. Moreover, early treatment with tocilizumab, an IL-6-blocking monoclonal antibody, was shown to augment myocardial salvage in patients with acute MI in other randomized trials [15,16,17]. Some deleterious predictors, such as high white blood cell and neutrophil counts, were identified, leading to adverse cardiovascular events in acute coronary syndromes, while others have been shown to be protective, such as anti-inflammatory responses mediated by CD11b- B1 lymphocytes affecting the infarcted muscle [18,19]. Conversely, the use of low-dose methotrexate failed to reduce either cardiovascular events or the serum levels of CRP and IL-6 in patients with stable atherosclerosis. This study provides important insights to investigators that a much more complicated process of inflammation underlies the pathogenesis of cardiovascular disease, and targeting non-specific steps of inflammation may not influence clinical outcomes [5]. Targeting systemic inflammation may have potential drawbacks, such as increasing the risk of infection or interfering with normal immune responses. Previous trials treating with corticosteroids have shown an increased incidence of ventricular arrhythmias, delayed healing and collagen deposition, and an extended infarct size [6,20]. However, targeting the specific steps of inflammation within the infarcted myocardium may be beneficial for patients with MI who have localized inflammation within the infarcted tissue. Recently introduced therapies targeting the recruitment or polarized macrophages within the infarcted tissue, such as antibodies against chemokines or the cytokines involved in these processes (e.g., targeting the CCL2/CCR2 axis), have shown promising results in preclinical studies [10,21,22].

## 3. Macrophages in Myocardial Infarction in Classical View

Resident cardiac macrophages play a housekeeping role in cardiomyocytes, such as promoting coronary artery development or maintaining tissue homeostasis [23]. Macrophages derived from monocytes migrate and infiltrate into the myocardium and seem to play a major role in tissue damage and destruction [24]. Immediately after MI, the proportion of resident cardiac macrophages drops in response to an ischemic stimulus, while monocyte-derived macrophages quickly replace those populations within a few days [25]. Differences in the expression profile of chemokine receptor type 2 (CCR2) account for this phenomenon, as its corresponding ligand, chemokine ligand 2 (CCL2), which is also called MCP-1, plays an important role in the migration of monocytes/macrophages. Monocyte-derived macrophages that extravasate from blood vessels to the myocardium transform into CCR2^+^ cardiac-resident macrophages [26]. Interactions with the local microenvironment, including damaged tissue or oxidative stress, and hypoxia, are associated with monocyte imprinting in MI [27]. An acute ischemic insult promotes the recruitment of monocytes to the myocardium through CCR2-MCP1-mediated trafficking and the secretion of proinflammatory mediators, such as IL-1, TNF, and IL-6 [28]. The changes in the phenotype of circulating CD14^+^ monocytes after MI also reflect the mobilization of monocytes from bone marrow and recruitment [29]. Human monocytes have been broadly categorized based on their surface expressions of CD14 and CD16. Classical monocytes (CD14^hi^ CD16^-^) are associated with several diseases hallmarked by inflammation and play an important role in the initiation and progression of the inflammatory response [30]. Nonclassical (CD14^low^ CD16^hi^) monocytes, in contrast, express high levels of CX3CR1 and low levels of chemokine receptors, including CCR2^-^ and Ly6C^low^, and they play a role in the resolution of inflammation [31].

A murine model with MI showed the penetration of distinct subsets of monocyte-derived macrophages: inflammatory Ly-6C^hi^ macrophages dependent on CCR2/CCL2 signaling in the earlier phase and regenerative Ly-6C^low^ macrophages via CX3CR1 signaling in the later phase. Classically, macrophages, defined as the M1 phenotype, infiltrate on days 1 to 3 after MI and promote the phagocytosis of necrotic cells by activating inflammation, while those with the M2 phenotype on days 5–7 after MI are responsible for the reconstitution of infarcted tissue and the resolution of inflammation (Figure 1) [32]. Dendritic cells are potent antigen-presenting cells that can activate and regulate T-cell responses [33]. T cells, particularly CD8^+^ cytotoxic and effective T cells, stimulate M1 macrophages and other proinflammatory pathways, while some effective T, CD4^+^, and regulatory T cells infiltrate the heart to promote M2 macrophages [33,34]. The use of the scRNA-seq technique in recent years has introduced various distinct transcriptomes among macrophage populations recruited from MI. For instance, some gene knockout studies distinguish diverse cell subsets of transcriptomes, and they do not necessarily align with one another [35].

## 4. Novel Concepts of Macrophages Revealed Using Single-Cell RNA Sequencing

Recent studies have utilized single-cell transcription sequencing techniques to distinguish and analyze heterogeneous cell populations by assessing the transcriptome at the single-cell level [9]. Prior studies of cardiac gene expression were limited to analyses at the whole-tissue level, which could not detect cell-specific changes or discriminate the origin of cells. The scRNA-seq technique provides an opportunity to detect disease-specific cell subsets or cell interactions critical in pathogenesis. Thus, these sequencing techniques systematically discover cell atlases and their heterogeneity in specific tissues and track dynamic molecular events in the progression of the disease.

The sequencing technique typically involves consecutive steps of single-cell isolation capture, cell dissolution, RNA reverse transcription, the amplification of complementary DNA, library preparation, sequencing, and a data analysis [36]. The development of analyses for data from the scRNA-seq method has been accompanied by advances in computational techniques. The pseudo-time analysis is one of the key techniques in single-cell research, which involves the sequencing of single cells along trajectories consistent with the similarity of gene expression patterns across sequenced cells to trace the process of dynamic cell change. Lineage tracing also provides comprehension in tissue development, the steady-state maintenance of homeostasis, and pathophysiology in certain diseases. The use of single-cell transcriptomics in combination with lineage tracing allows for the investigation of the transcriptional state of thousands of individual cells, enabling a reliable analysis of the diversity of cell types and transitions between distinct states in diverse samples [37].

Table 1 lists recent animal studies that use scRNA-seq to investigate the acute cellular response after MI. Vafadarnejad et al. observed diverse neutrophil subsets, which are basically considered first responders to tissue damage, over time in a murine MI model [38]. Mildner et al. demonstrated heterogeneity in the transcriptional profiles of monocytes with intermediate expression levels of Ly-6C other than Ly-6C^hi^ and Ly-6C^low^, supporting the notion that these cells represent an intermediate developmental stage of monocytes [39]. Likewise, diverse and distinct transcriptomes of recruited macrophages in MI have been elucidated, and the canonical M1–M2 polarized paradigm is accepted as an oversimplified model [40]. Zhuang et al. observed that macrophages from various origins had diverse transcriptional signatures, transcriptional modulators, or pathways and showed distinct developmental trajectories through scRNA-seq [41]. Bajpai et al. recently identified one monocyte group and seven distinct macrophage groups for cell clustering based on expression markers in a mouse MI model [28]. This study confirmed the heterogeneity of macrophages in the heart after MI, especially how CCR2^+^ and CCR2^-^ macrophages mediate inflammatory responses (Figure 2). Interestingly, depleting CCR2^+^ macrophages prior to the induction of MI allowed fewer monocytes to be recruited into the myocardium, while depleting CCR2^-^ macrophages resulted in an increase in the number of recruited monocytes. Ni et al. also investigated cardiac macrophage heterogeneity and identified a subset of macrophages associated with myocardial damage using an scRNA-seq analysis. CD72^hi^ cardiac macrophages were found to be a proinflammatory macrophage subset [42]. This in vivo complexity revealed through scRNA-seq techniques provides recent insights into macrophage biology and may explain the failure of nonselective immunosuppressive strategies in MI. At the same time, it offers numerous novel cellular and molecular targets for investigators to modulate LV remodeling following MI.

## 5. Trem2^hi^ Macrophage in Cardiovascular Disease and Myocardial Infarction

Recently, Jung et al. used scRNA-seq to identify dominant subsets of neutrophils, monocytes, and macrophages over time after MI in mouse models [10]. They investigated longitudinal spatial transcriptome profiles from mouse hearts, from day 1 to day 7 after inducing MI, to put unbiased gene expression profiles together in corresponding tissue sections, providing information on the infiltrative feature of immune cells (Figure 2). These analyses with the deconvolution algorithm allowed us to trace the proportions of monocytes, neutrophils, and macrophages in a time-dependent manner, and our findings are consistent with those in previous reports [28,42]. Moreover, from a sub-clustering analysis in the context of scRNA-seq data, these cells were concentrated in the infarcted area in different manners, with neutrophils infiltrating into the infarcted area during early MI and macrophages infiltrating into the infarcted area during subacute MI, demonstrating the dynamic recruitment of immune cells [9,10]. A single-cell trajectory analysis provided support for sequential differentiation from Ly-6C^hi^ monocytes to late macrophages with upregulated Trem2 gene expression in a time-dependent manner. Gene expression in macrophage subsets confirmed with scRNA-seq was dynamically regulated. Early macrophage-specific genes (*Ccr2*, *Chil3*, and *Clec4e*) were upregulated in Ly-6C^hi^ monocytes and early macrophages, while their expressions gradually decreased to low levels in late macrophage populations (Figure 1) [10]. Conversely, late-macrophage-specific genes (*Trem2*, *Rgs10*, and *Fcrls*) were downregulated in Ly-6C^hi^ monocytes and early macrophages but gradually upregulated in late macrophages during the course of MI. In particular, Trem2 was particularly upregulated in late-macrophage subsets rather than in early subsets (Figure 3).

Trem2 is a transmembrane receptor of the immunoglobulin superfamily, and it propagates signaling through interaction with the adaptor proteins DNAX activation protein (DAP) 10 and DAP12. Trem2 binds to various molecules and interacts with other signaling pathways in complicated manners. Its downstream signaling may represent a danger signal in tissue damage or disease, while it is known to have a physiological role in tissue development or maintenance in specific microenvironments, such as those of the brain or bone. Its biological role is known to significantly modify cellular functions via the induction of phagocytosis, the restriction of inflammation, and the promotion of macrophage survival [46,47].

Previous studies investigating the association between Trem2 and heterogenous diseases, including cardiovascular disease, are depicted in Table 2. Trem2 has been studied in Alzheimer’s disease, as its activation plays a role in the formation of pathologic β-amyloid, as well as that of lipoproteins and apolipoproteins [48,49]. In cancer immunology, accumulating evidence suggests a pathological role of Trem2 in establishing an Immune-suppressive tumor niche in malignancies. M. Molgora et al. adopted scRNA-seq to reveal that the deletion of Trem2 accompanies an improvement in T-cell responses. They also found the expression of Trem2 in tumor-associated macrophages in over 200 human cancers, and some were inversely correlated with prolonged survival [50]. Associated with obesity, Jaitin et al. observed that Trem2 activation induces apolipoproteins in adipose tissue with obese mice models [51]. Other murine model research demonstrated that the neutralization of Trem2 resulted in the inhibition of adipocyte differentiation in vitro and in vivo through the downregulation of the expressions of adipogenic regulators [52]. In liver cirrhosis or fatty liver, Trem2 is regarded as a pro-fibrotic marker associated with scar formation. Ramachandran et al. identified the expansion of Trem2^+^ CD9^+^ macrophages in human fibrotic livers [53].

In cardiovascular disease, recent studies support the role of Trem2 in atherosclerosis. Atherosclerotic plaque is formed by accumulating fat-rich macrophages due to the recruitment of circulating monocytes and the differentiation and proliferation of local macrophages [54]. Recently, it has been reported that Trem2^hi^ macrophages increase lipid metabolism in atherosclerotic plaque. Moreover, a high expression of Trem2 can reduce the stability of plaque. In Trem2^hi^ macrophages, researchers observed increased levels of cathepsins, which could increase plaque vulnerability and the inflammation of atherosclerosis [55]. Trem2 may also contribute to the microcalcification of atherosclerotic lesions that can worsen local tissue stress and impair plaque stability [56,57].

A recent scRNA-seq study in a murine model distinguished subsets of Trem2^hi^ macrophages in the mouse heart during the progression of atherosclerosis, which did not align with the M1 or M2 profile [58]. However, its infiltrative behavior and impact on MI healing have never been investigated. Fu et al. demonstrated that Trem2 expression was upregulated in MI-induced mice, possibly through the activation of the PI3K/AKT signaling pathway. A mice model with MI has shown that Trem2-transfected mice were associated with an alleviated injury, a reduced infarct size, and a decreased number of apoptotic cells, as well as with better echocardiographic function [59]. Some other studies provided a comprehensive landscape of cardiac macrophage states in the acute phase of MI by using a scRNA-seq analysis and focused on the regulatory function of Trem2. Recently, Jung et al. conducted their research to demonstrate whether Trem2^hi^ macrophages infiltrate specific areas or time points post-MI using concomitant analyses [10]. They noted pro- and anti-inflammatory gene expression activities in such cells to identify the anti-inflammatory characteristics of Trem2^hi^ macrophages. They orthogonally validated the expression of Trem2 in the heart after MI through immunohistochemistry, Western blotting, co-localization assays, flow cytometry, and quantitative RT–PCR (Figure 2 and Figure 3). Trem2^hi^ macrophages specifically accumulated in the infarcted area of the myocardium in the subacute period of MI, expressing a rich set of anti-inflammatory genes, cytokines, and chemokines, such as *Arg1*, *Alox15*, *TGF-β*, *IL-4*, *IL-10*, *Ccl1*, and *Cxcr1*.

**Table 2 ijms-24-05556-t002:** Summary of previous studies on the role of triggering receptor expressed on myeloid cells 2 (TREM2) in specific diseases, including cardiovascular disease.

Study	Disease	Main Findings
Yeh et al. (2016) [49]	Alzheimer	The uptake of low-density lipoprotein and apolipoproteins E and J was increased in heterologous cells when Trem2 was overexpressed. On the contrary, disease variants in Trem2 impaired lipoprotein binding and uptake into cells.
Ulland et al. (2017) [48]	Alzheimer	The risk of developing Alzheimer’s disease is increased in individuals with hypomorphic variants of Trem2. In mouse models, Trem2-deficient microglia display autophagy.
Park et al. (2015) [52]	Obesity	In Trem2 transgenic mice fed with a high-fat diet, the upregulation of adipogenic regulators was observed, which indicates that Trem2 promotes obesity and adipogenesis.
Jaitin et al. (2019) [51]	Obesity	Using scRNA-sec, researchers observed a dynamic adipose tissue immune cell atlas in both mice and humans. The study revealed that Trem2 plays a key role in the protective functions of LAMs, which counteract inflammation, adipocyte hypertrophy, and metabolic disease.
Ramachandran et al. (2019) [53]	Liver cirrhosis	Trem2 has been identified as a profibrotic marker associated with scar formation. In this study on human liver cirrhosis, researchers were able to resolve the fibrotic niche and identify the association of Trem2^+^ CD9^+^ macrophages.
Cochain et al. (2018) [58]	Atherosclerosis	The identification of arterial leukocytes in a murine atherosclerotic aorta model revealed a subset that is associated with atherosclerosis and expresses Trem2.
Depuydt et al. (2020) [56]	Atherosclerosis	The researchers identified a foam-cell-like myeloid cell population expressing Trem2 in human carotid atherosclerotic plaques.
Waring et al. (2022) [57]	Calcification	Trem2^hi^ macrophages generate microcalcifications in plaques and are associated with plaque instability in mice.
Fu et al. (2022) [59]	Myocardial infarction	The expression of Trem2 was found to be increased in mice with myocardial infarction, and it was observed that Trem2 may activate the PI3K/AKT signaling pathway. In Trem2-transfected mice, reduced injury, smaller infarct size, and fewer apoptotic cells were observed, as well as better echocardiographic function.
Jung et al. (2022) [10]	Myocardial infarction	After myocardial infarction in mice, there was an increase in Trem2 expression in macrophages during the subacute phase of cardiac injury. Administering soluble Trem2 in vivo resulted in marked enhancements in both the functional and structural aspects of the infarcted hearts.

scRNA-seq defined as single-cell RNA sequencing; LAM, lipid-associated macrophage.

## 6. The Role of Trem2 as a Therapeutic Target and Future Direction

Monocytes and macrophages are known to play key roles in ventricular remodeling after MI. Nevertheless, the effectiveness of targeted immunotherapy strategies needs to be improved, and strategies sharing specific targets show conflicting results. Many recent preclinical studies have also focused on modifying monocyte responses or targeting the CCL2/CCR2 axis to suppress monocyte-trained immunomodulators after MI inflammation [21,22]. Applying the anti-inflammatory effect of the Trem2 gene could be a new therapeutic strategy for treating MI [60]. The classical approach to obtaining the beneficial effects associated with TREM2 macrophage function is to target the signaling pathways that regulate Trem2 expression and activation. However, the receptor of Trem2 has not yet been revealed, so it cannot be targeted. Recently, an alternative method using a soluble TREM2^hi^ macrophage injection technique has been introduced. Jung et al. demonstrated the increasing expression of the soluble form of Trem2 (sTrem2, 18 kDa) beginning on day 3 post-MI and peaking on day 7. They injected sTrem2 mixed with a gelatin hydrogel (sTrem2-GH) in vivo and identified that sTrem2 promotes functional and structural improvements in damaged hearts in mice (Figure 4). At 28 days after MI, the mice treated with sTrem2-GH showed a histologically smaller infarct size and a better echocardiographic LV systolic function, verifying its in vivo effect on LV remodeling after MI [10]. Moreover, overall survival was superior in the mice treated with sTrem2-GH compared to that in the control mice.

Despite the identification of Trem2^hi^ macrophages as a possible modulator in cardiac remodeling after MI, further studies are needed to evaluate their systematic properties before conducting human clinical trials. The origin of Trem2^hi^ macrophages, sets of cells affected by sTrem2, and interacting intracellular signaling cascades should be elucidated to allow for a better understanding of their regenerative effects in infarcted hearts. Additionally, the molecular interactions and ligands of Trem2 in human macrophages need to be compared and evaluated before targeting Trem2^hi^ macrophages in humans. Furthermore, potential side effects should be considered, such as impaired immune responses, unintended off-target effects on other signaling pathways or cellular functions, and potential neurodegenerative effects, as Trem2 is also expressed in the central nervous system and is involved in microglial function regulation.

## 7. Conclusions

The spatiotemporal dynamics of immune cells during the acute phase of post-MI have recently been studied, shedding light on their complex interplay. This new understanding shifts the paradigm of myocardial cellular composition after MI and provides potential targets for MI treatment. Among the heterogeneous macrophage populations, Trem2^hi^ macrophages show promise due to their protective inflammatory properties. In an in vivo experiment with mice, a soluble Trem2 injection during the subacute phase of MI significantly improved myocardial function. Further investigation into the role of Trem2^hi^ macrophages in the post-MI immune response and the regulatory signaling of Trem2 will aid in the development of novel therapeutic strategies for MI.

## Figures and Tables

**Figure 1 ijms-24-05556-f001:**
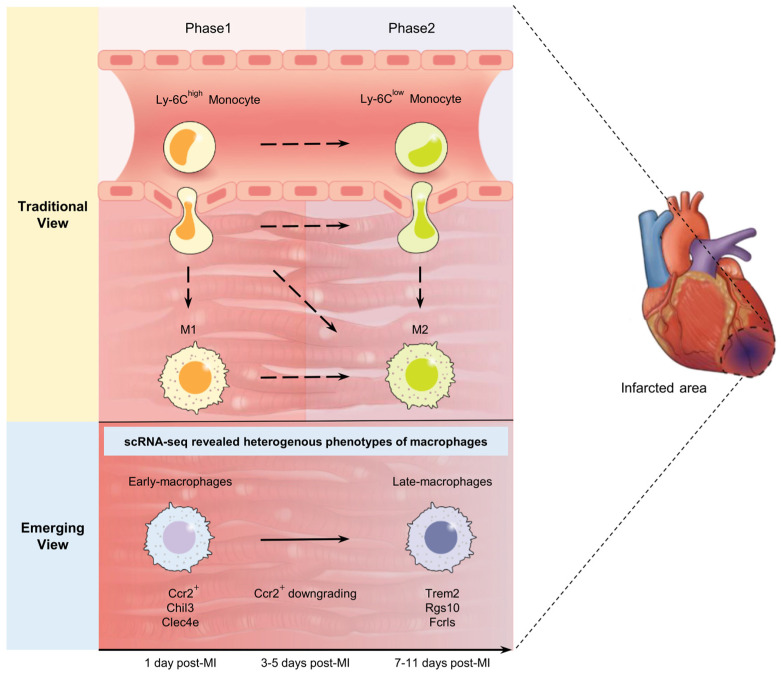
Traditional and emerging concepts of monocytes/macrophages in infarcted myocardium. The traditional concept defines the dichotomous classification of Ly-6C2^hi^/Ly-6C2^low^ monocytes and M1/M2 macrophages by the temporal phase. Inflammatory Ly-6C2^hi^ monocyte-derived M1 macrophages reside in the earlier phase, while regenerative Ly-6C2^low^ monocytes and derived M2 macrophages reside in the later phase. An emerging concept discriminates the heterogeneous transcriptomic genotypes of monocytes and macrophages in infarcted myocardium. Macrophage subsets with a heterogeneous spectrum of gene expression suggest the differentiation or dynamic interchange of immune cells in the course of myocardial infarction.

**Figure 2 ijms-24-05556-f002:**
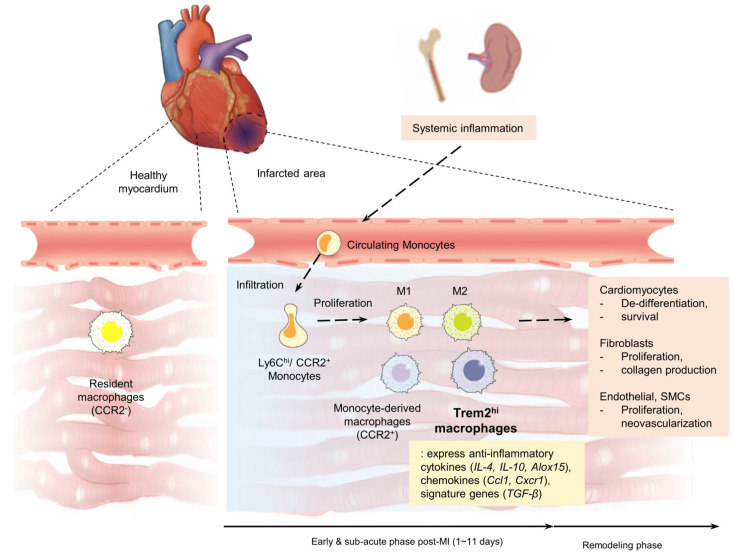
The contribution of circulating monocyte infiltration and monocyte-derived macrophages, as well as soluble Trem2 as a potential modulator of cardiac remodeling, in myocardial infarction. During a myocardial infarction, the bone marrow and spleen become instrumental in the delivery of monocytes that infiltrate the infarcted myocardium [43]. Upon arrival, these monocytes differentiate into macrophages and subsequently undergo proliferation. The infarcted myocardium recruits a significant number of leukocytes, including circulating inflammatory Ly-6C^hi^ monocytes, which are guided by the chemokine–chemokine receptor pair CCL2-CCR2 to infiltrate the site of injury. These monocyte-derived macrophages contribute to inflammation by secreting cytokines, proteases, and oxidative-stress-promoting enzymes [27]. After arriving in the infarcted myocardium, monocytes differentiate into macrophages, which can proliferate locally and persist for weeks alongside reparative embryonic macrophages. These monocyte-derived macrophages contribute to efferocytosis, cardiac myocyte survival, angiogenesis, and fibrosis [44,45]. In a recent preclinical study utilizing scRNA-seq, Jung et al. found evidence supporting the modulation of cardiac remodeling by the increasing expression of the soluble form of Trem2 during the 3–7 day period post-MI [10].

**Figure 3 ijms-24-05556-f003:**
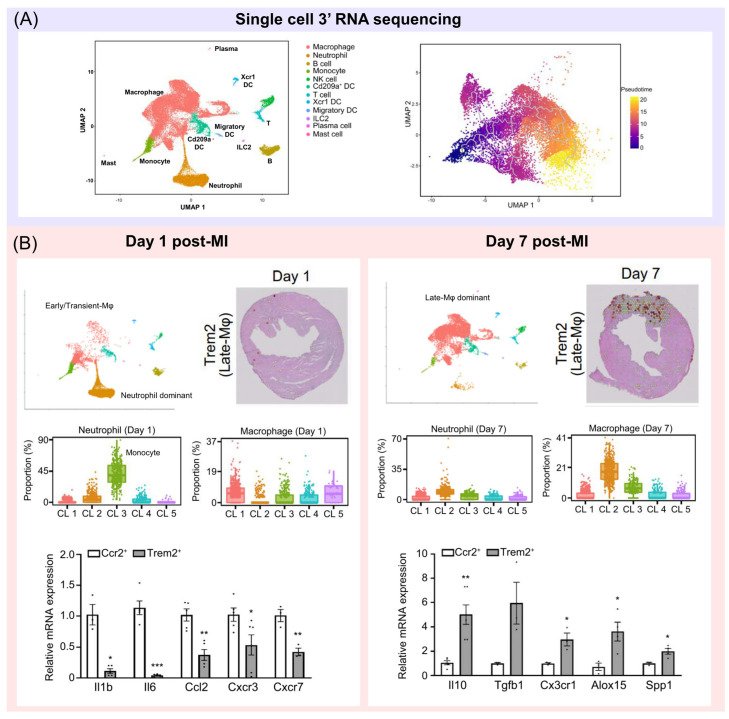
The dynamics of immune cells over time and the presence of Trem2^hi^ macrophages in mouse hearts following myocardial infarction [10]. (**A**) Time-dependent immune cell dynamics visualized in two-dimensional uniform manifold approximation and projection. Each point represents a single cell, and cell types are color-coded. The results of single-cell trajectory analysis showed a sequential differentiation process from monocytes to late macrophages. (**B**) The proportion of neutrophils and macrophages among the total immune cells infiltrated into the myocardial tissue changes over time after MI. The early phase (day 1) is dominated by neutrophils and monocytes, whereas the subacute phase (day 7) shows a population dominated by macrophages, with high expression of the Trem2 gene, as demonstrated by spatial transcriptome sequencing. The mRNA expression levels of pro- and anti-inflammatory genes were compared between early macrophages (day 1 post-MI) and late macrophages expressing Trem2 (day 7 post-MI), revealing an upregulation of anti-inflammatory genes in Trem2^+^ late macrophages. Unpaired two-tailed t-test was used to determine the statistical significance. * *p* < 0.05, ** *p* < 0.01, *** *p* < 0.001.

**Figure 4 ijms-24-05556-f004:**
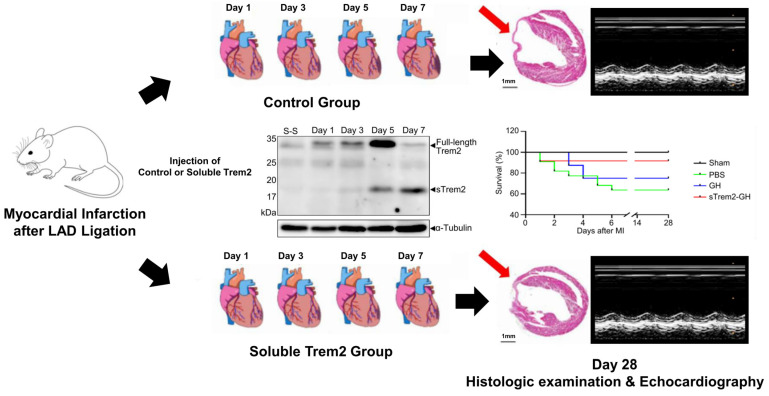
In vivo function of soluble Trem2 in mouse hearts after MI [10]. Left ventricle function and survival following myocardial infarction were assessed, comparing the effect of sTrem2. Mice with ligated left anterior descending arteries were injected with either control or sTrem2 molecule. Echocardiographic left ventricle function and histologic infarct size were compared 28 days post-MI, showing a smaller infarct size and better echocardiographic function in sTrem2 group; 28 days survival was superior in sTrem2-injected group.

**Table 1 ijms-24-05556-t001:** Summary of previous studies applying single-cell RNA sequencing associated with acute cellular response after myocardial infarction.

Study	Study Design	Model	Cell Types	Main Findings
Vafadarnejad et al. (2020) [38]	Animal research	Murine/heart after MI	Neutrophil	While circulating neutrophils undergo an aging process characterized by the loss of surface CD62 ligand and upregulation of *Cxcr4*, it has been shown that cardiac-infiltrating neutrophils acquire a unique SiglecF^hi^ signature.
Mildner et al. (2017) [39]	Animal research	Murine monocyte isolation from blood	Monocyte	The use of scRNA-seq and transplantation experiments has confirmed that Ly-6C^int^ monocytes are a heterogeneous population that expresses MHCII. The differentiation of Ly-6C^+^ monocytes into Ly-6C^-^ cells is regulated by the transcription factor C/EBPβ. C/EBPβ is also involved in the regulation of *Nr4a1* expression during monocyte conversion.
Zhuang et al. (2020) [41]	Animal research	Mouse/MI surgery	Macrophage, myofibroblast	Macrophages from different origins exhibited divergent transcriptional signatures, pathways, developmental trajectories, and transcriptional regulons. Myofibroblasts predominantly expand at 7 days after myocardial infarction with pro-reparative characteristics (signature genes: *Postn, Cthrc1, and Ddah1, among Ddah1*).
Bajpai et al. (2019) [28]	Animal research	Cardiomyocytes of mouse MI model	Monocyte, macrophage	Identification of heterogeneity of macrophages in the heart after MI (1 monocyte cluster and 7 clusters of macrophages). A set of cardiac-resident CCR2^+^ macrophages are responsible for the recruitment of monocytes to the injured myocardium. Tissue-resident CCR2^-^ macrophages oppose the recruitment.
Ni et al. (2021) [42]	Animal research	Murine models of HF (transverse aortic constriction and chronic Ang II infusion)	Cardiomyocyte, fibroblast	Cardiac macrophage heterogeneity was identified using scRNA-seq analysis, which revealed a subset of macrophages associated with heart damage. Additionally, a proinflammatory macrophage subset, identified as CD72^hi^ cardiomyocytes, was also discovered.
Jung et al. (2022) [10]	Animal research	MI-induced mice	Neutrophil, monocyte, macrophage	Identification of major immune cell populations and their proportion changes after MI over time. The upregulation of Trem2 expression in macrophages was observed with trajectory inference analysis during the subacute phase of acute MI. Infarcted hearts were significantly improved by in vivo injection of soluble Trem2.

MI defined as myocardial infarction; scRNA-seq, single-cell RNA sequencing; Ang, angiotensin; HF, heart failure.

## Data Availability

No new data were created.

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
