# Peer review of "The Protective Role of TREM2 in the Heterogenous Population of Macrophages during Post-Myocardial Infarction Inflammation"

_ijms, 2023, doi:10.3390/ijms24065556_

Round 1

Reviewer 1 Report

Reviewer’s Comments:

The manuscript “Heterogeneity of macrophages and the protective inflammatory role of TREM2 during the post myocardial infarction phase” is a very interesting work. In this work, Advances in interventions after myocardial infarction (MI) have dramatically increased survival, but MI remains the leading cause of heart failure due to maladaptive ventricular remodeling following ischemic damage. Inflammation is crucial in both the initial response to ischemia and subsequent wound healing in the myocardium. To date, preclinical and clinical efforts have been made to elucidate the deleterious effects of immune cells contributing to ventricular remodeling and to identify therapeutic molecular targets. The conventional concept classifies macrophages or monocytes into dichotomous populations, while recent studies support their diverse subpopulations and spatiotemporal dynamicity. While I believe this topic is of great interest to our readers, I think it needs major revision before it is ready for publication. So, I recommend this manuscript for publication with major revisions.

1. In this manuscript, the authors did not explain the importance of the protective inflammatory the introduction part. The authors should explain the importance of protective inflammatory.

2) Title: The title of the manuscript is not impressive. It should be modified or rewritten it.

3) Correct the following statement “Trem2hi macrophages possess an anti-inflammatory nature and express soluble Trem2, which contributes to improved Left ventricle (LV) function in vivo”.

4) Keywords: The protective inflammatory is missing in the keywords. So, modify the keywords.

5) Introduction part is not impressive. The references cited are very old. So, Improve it with some latest literature like 10.3390/molecules27217368, 10.3390/molecules27196580

6) The authors should explain the following statement with recent references, “Moreover, these cells were concentrated in the infarcted area in different manners, with neutrophils infiltrating into the infarcted area during early MI and macrophages infiltrating into the infarcted area during late MI, demonstrating dynamic recruitment of immune cells to the injured heart tissues after MI”.

7) Add space between magnitude and unit. For example, in synthesis “21.96g” should be 21.96 g. Make the corrections throughout the manuscript regarding values and units.

8) The author should provide reason about this statement “In addition to their anti-inflammatory properties, Trem2hi macrophages expressed sTrem2 in the subacute phase of MI, which significantly improved myocardial function and remodeling of the infarcted heart in vivo (Figure 3)”.

9. Comparison of the present results with other similar findings in the literature should be discussed in more detail. This is necessary in order to place this work together with other work in the field and to give more credibility to the present results.

10) Conclusion part is very long. Make it brief and improve by adding the results of your studies.

11) There are many grammatic mistakes. Improve the English grammar of the manuscript.

Reviewer 2 Report

The manuscript entitled, “Heterogeneity of macrophages and the protective inflammatory role of TREM2 during the post myocardial infarction phase" by Kim et al., studied the role of inflammation in both the initial response to ischemia and subsequent wound healing in the myocardium. In this review authors complied major studies to characterize the dynamics of macrophages during the acute period of MI and to investigate these underlying inflammatory processes and the spatiotemporal dynamics of MI-associated immune cells. Additionally, they introduce Trem2hi macrophages and the soluble form of Trem2, a major anti-inflammatory immune cell in infarcted myocardial and its essential molecule, resulting in notable functional and structural improvements in infarcted hearts.

While this is a well-written, important, and timely article, there are some changes this reviewer would make to make for a more complete examination of the subject:

Comments:

1 The English of the manuscript could be refined (minor), and there are few typographical errors.

2, The figures 1, 2A, and 3 can be enhanced (high resolution).

3, In order to emphasize the study's summary, at least one graphical illustration may be included.

4, The authors must double-check every abbreviation in the article. First, define using the full name, then the abbreviation.

5, Authors should add a paragraph or few lines about immune cells specifically adaptive immune cells such as CD4 and CD8 in this review (PMID: 16751419, PMID: 36093172, PMID: 36465455, PMID: 36337927; PMID: 34630414; PMID: 34043424; PMID: 35730443 etc). This can broaden the article's perspective and make it more useful to a wider audience.

Reviewer 3 Report

In this review the authors discussed diverse macrophage subpopulations and spatiotemporal dynamicity in infarcted hearts. They proposed that Trem2hi macrophages were recruited to infarcted myocardial tissue in the late phase of myocardial infarction, playing an anti-inflammatory role in improving left ventricle remodeling. The topic is interesting. Some concerns and suggestions are listed as below:

Regarding the role of inflammation in myocardial infarction, the evidece supported that systemic inflammation plays a key role in the pathogenesis of myocardial infarction. Potential mechanisms can be summaried using a figure. When (early or late phase) inflmmation should be targeted? How about the relationship between systemic inflammation and tissue macrophages in myocardial infarction (since you metioned tissue macrophages later)? 

The authors said that targeting non-specific steps of inflammation may not influence on clinical outcomes (in lines 94-95). No need to target systemic inflammation? Potential reasons should be discussed in details.

In Figure 1, the authors discussed Ly6C monocytes in myocardial infarction. Can monocytes be imprinted by the microenvironment? Which factors? How about changes of CD14+ monocytes in human?

In lines 152-153, no need to mention neurophils.

In the part of novel concepts of macrophages revealed by single cell RNA sequencing, I did not find any novel concepts. The description was too general. The biggest issue is that the review is simply a repetition of the literature with no attempt to synthesize or critically discuss the results presented. 

The page numbers (upper corner) are not correct in this paper. 

I wonder if Trem2hi macrophages have been validated in patients with myocardial infarction.

How Trem2 can be targeted for beneficial effects? Potential methods should be discussed and summarized.

Any potential side effects by targeting Trem2?

Round 2

Reviewer 3 Report

The authors have addressed my concerns.